# Variation between hospital caesarean delivery rates when Robson's classification is considered: An observational study from a French perinatal network

Thibaud Quibel[1,2]*, Patrick Rozenberg[1,2], Camille Bouyer[1,3], Jean Bouyer[4]

1 Department of Gynecology and Obstetrics, Intercommunal Hospital Centre of Poissy-Saint-Germain-en-Laye, Poissy, France, 2 EA 7285, Research Unit "Risk and Safety in Clinical Medicine for Women and Perinatal Health", Versailles-Saint-Quentin University (UVSQ), Montigny-le-Bretonneux, France, 3 Réseau de Périnatalité Maternité en Yvelines et Périnatalité Active (MYPA), Saint-Germain-en-Laye, France, 4 Université Paris-Saclay, UVSQ, Inserm, CESP, Villejuif, France

* thibaud.quibel@ght-yvelinesnord.fr

**Data Availability Statement:** All relevant data are within the manuscript and its S1 File.

**Funding:** The authors received no specific funding for this work.

## Abstract

### Introduction

WHO has recommended using Robson's Ten Group Classification System (TGCS) to monitor and analyze CD rates. Its failure to take some maternal and organizational factors into account, however, could limit the interpretation of CD rate comparisons, because it may contribute to variations in hospital CD rates.

### Objective

To study the contribution of maternal socioeconomic and clinical characteristics and hospital organizational factors to the variation in CD rates when using Robson's ten-group classification system for CD rate comparisons.

### Methods

This prospective, observational, population-based study included all deliveries at a gestational age > 24 weeks at the 10 hospitals of the French MYPA perinatal network in the Paris area. CD rates were calculated for each TGCS group in each hospital. Interhospital variations in these rates were investigated with hierarchical logistic regression models to quantify the variation explained by differences in patient and hospital characteristics when the TGCS is considered. Variations in CD rates between hospitals were estimated with median odds ratios (MOR) to express interhospital variance on the standard odds ratio scale. The percentage of variation explained by TGCS and maternal and hospital characteristics was also calculated.

**Competing interests:** The authors have declared that no competing interests exist.

## Results

The global CD rate was 24.0% (interhospital range: 17–32%). CD rates within each TGCS group differed significantly between hospitals ($P<0.001$). CD was significantly associated with maternal age (>40 years), severe preeclampsia, and two organizational factors: hospital status (private maternities) and the deliveries per staff member per 24 hours. The MOR in the empty model was 1.27 and did not change after taking the TGCS into account. Adding maternal characteristics and hospital organizational factors lowered the MOR to 1.14 and reduced the variation between hospital CD rates by 70%.

## Conclusion

Maternal characteristics and hospital factors are needed to address variation in CD rates among the TGCS groups. Therefore, comparisons of these rates that do not consider these factors should be interpreted carefully.

## Introduction

The cesarean delivery (CD) has become the most commonly performed operation around the world [1]. Despite the 1985 recommendation of the World Health Organization (WHO) that "[t]here is no justification for any region to have a cesarean delivery rate higher than 10–15%" [2], CD rates have continued to increase worldwide and have led to major and controversial public health problems [3, 4]. Although CDs remain essential for some obstetric conditions (for example, placenta previa), many studies have highlighted harmful consequences of its rise, especially the notably higher risks for mothers and babies in subsequent pregnancies, such as abnormally invasive placenta, placental abruption, and stillbirth [5–8]. Consistent with the lack of clear evidence about the optimal mode of delivery, CD rates vary considerably between institutions, regions, and countries [9–12]. These differences may be due to many factors including variations in patient characteristics or preferences, access to care, clinician behavior, and hospital culture or policy [13–17].

Several classification systems have been devised to analyze and try to understand discrepancies in CD rates between hospitals [18–20]. In 2011, a systematic review of 27 CD classification systems identified the 10-group classification system (TGCS) proposed by Robson in 2001 as the most appropriate for classifying women according to their CD risk [21]. The TGCS classifies all deliveries into one of 10 groups based on 6 variables: parity, history of CD, mode of onset of labor, fetal presentation, multiple pregnancy, and gestational age. These 10 categories are mutually exclusive, totally inclusive and can be applied prospectively, since each woman admitted for delivery can be classified immediately according to a few routinely recorded variables [22]. In 2015, WHO recommended that healthcare facilities use the TGCS as a common starting point for comparing and analyzing CD rates (between hospitals), in the expectation that this classification would help healthcare providers to develop interventions to optimize CD rates. Indeed, its use implicitly suggests that women in each group have the same risk profile for CDs. However, the downside of the simplicity of this classification is that it does not take into account many other risk factors related to maternal characteristics, including but not limited to maternal age, obesity, diseases such as hypertensive disorders, and hospital organizational factors, all of which can also contribute to variations in CD rates. Therefore, the use of the single TGCS classification may lead to inappropriate conclusions in the comparison of CD rates between healthcare facilities.

The aim of this study was to examine the contribution of maternal socioeconomic and clinical characteristics and hospital organizational factors to the variation in CD rates when considering Robson's ten-group classification system.

## Material and methods

This study used data from a population-based cohort of births which occurred between the first January 2014 and the 31st December 2014 in in the MYPA (Maternité en Yvelines et Périnatalité Active) perinatal network, which covers the district of Yvelines (west of Paris). This network, which manages around 18,000 deliveries per year, is composed of 10 hospitals (referred to as hospitals A to J), including one hospital (A) affiliated with a university (Université Versailles-Saint Quentin). Half of the hospitals were public (A, B, C, E, H) and half private (D, F, G, I, J). Only two hospitals (A and B), both of which include a neonatal intensive care unit (NICU), manage all types of pregnancies. The other eight (C to J) are level-1 departments intended to provide services only for low-risk pregnancies. The number of deliveries/years differed notably between hospitals: units A, B, and C each had more than 2,000 deliveries/year; units D, E, F, G, and H had between 1,000 and 2,000; and units I and J had fewer than 1,000.

Data are extracted from the CoNaissance 78 program, which was created in 2008 to monitor maternal and perinatal morbidity and mortality in the perinatal network. This dataset contains all births in the district with fully anonymized demographic characteristics and medical information about each pregnancy and delivery, as well as about maternal, fetal, and neonatal health. Therefore, any of the authors had access to data with identifying information prior to de-identification in the database. Since its creation, data are continuously recorded from two health certificates completed for each birth in the network at the hospital of delivery:

- the "first health certificate" of infants born in network hospitals, which is completed during a medical examination that is compulsory nationwide, to be performed within eight days after birth, usually in the maternity ward;

- an additional local health certificate, specifically developed within the MYPA network and reporting additional anonymized data including severe maternal morbidity, social characteristics (e.g., lack of health insurance, single parenthood), and data for all fetal deaths and medical terminations of pregnancy at and after 22 weeks of gestation.

These certificates are completed by midwives and physicians in each hospital and then collected and recorded. Two research midwives provide assistance in collecting the missing data for the CoNaissance 78 program database and in controlling its quality to improve the completeness of the certificates. The proportion of missing data is < 3% for all variables in this study [23]. The National Committee for Data Protection (Commission Nationale de l'Informatique et des Libertés, registration number 1295794) approved the study, which was conducted in accordance with French legislation. Under French law, this study is exempt from informed consent requirements because patients received standard care and because the dataset contains no information that enables patient identification. Similarly, ethics committee approval was not required because the study used an anonymized database and did not modify patient care.

The maternal risk profile included the following variables: maternal age, recorded and coded in two categories (<40 years yes/no), educational level (coded in three categories: university/high school/primary and middle school), occupational activity (yes/no), and severe preeclampsia, defined according to the ACOG criteria in the perinatal network [24]. Hospital organizational factors included hospital status (public or private), the presence of a neonatal intensive care unit (yes/no), and the ratio of deliveries to staff members (midwives and obstetricians) per 24 hours. This last variable was developed to create a quantitative measure for

appropriate staffing, in view of the lack of any standard for obstetric care in labor wards between countries, given that deliveries are managed by either midwives or obstetricians. In our analysis, we took the median of deliveries per staff member per 24 hours among MYPA hospitals for a reference.

For each TGCS group, we calculated its specific cesarean rate (that is, the number of women undergoing a CD divided by the total number of women in the group), and the relative size of the group (as a percentage of the total population of women within each hospital). We also calculated each group's contribution to the overall CD rate, also as a percentage, that is, the number of women with a CD in the group divided by the total number of CDs.

The variation in hospital CD rates was analyzed with a multilevel logistic regression model with a random intercept to take into account that women were nested within hospitals [25–27]. This model enabled us to estimate the area level variance to quantify the variation in CD rates between hospitals [28]. We then computed the median odds ratio (MOR) to express hospital-level variance according to the standard odds ratio (OR) scale, which can be interpreted consistently and intuitively. The MOR is defined as the median value of the odds ratio between the hospital at highest risk of CD and the hospital at lowest risk, with random selection of two hospitals. MOR can be conceptualized as the increased median risk that a patient would have if moved to another hospital with a higher risk. In this study, the MOR showed the extent to which the individual probability of a CD was determined by the hospital's level of care and was therefore appropriate for quantifying contextual phenomena [25]. If the MOR is equal to one, the probability of a CD does not differ between the hospitals.

Models were fitted with the three-stage approach described by Merlo et al. [25]. First, a null (or empty) model with hospital random intercepts only was fit to estimate the unadjusted hospital-level variation in CD rates. Then we added the TGCS group as a variable (that is, the group in which each woman is classified) in the model to investigate the extent to which the variation in CD rates between hospitals was modified when the TGCS classification was considered. Finally, we added the variables describing the women's clinical, social, and demographic characteristics as well as the hospitals' characteristics, to determine the amount of variation associated with these variables and to test the significance of the residual between-hospital variance. Maternal and hospital variables whose crude associations with CD occurrence had a p-value $< .2$ were selected for the multilevel model.

As more than 25% of deliveries took place in a single hospital (A, 4,530/17,433), we performed a sensitivity analysis by excluding it. Variation in CD rates among the nine remaining hospitals was again analyzed with the same process.

All analyses were performed with R Studio version 1.0.136. The analysis was conducted by applying a significance level of $\alpha = 0.05$.

## Results

During 2014, 17,511 deliveries took place in the 10 maternity units of the perinatal network. After excluding 78 women (0.4%) due to missing data for one of the 6 TGCS variables, 17,433 women remained in the final sample. Two thirds of the deliveries occurred in a public hospital (66.0%, 11,512/17,433), and 40.0% (6,999/17,433) in the hospitals with a NICU. There were 4,182 CDs during this year (overall rate: 24.0%, interhospital range: 17.2%-32.6%). Fig 1 shows the CD rate for each hospital, and Table 1 summarizes the characteristics of the women and the hospitals by TGCS group.

Most women were considered at low risk, i.e., classified in TGCS groups 1 and 3 (56.2%; 9,804/17,433), including nulliparous and parous women with no previous CD, with a singleton in cephalic presentation $\geq$ 37 weeks, with spontaneous labor. Groups 2 and 4 (4,003/17,433;

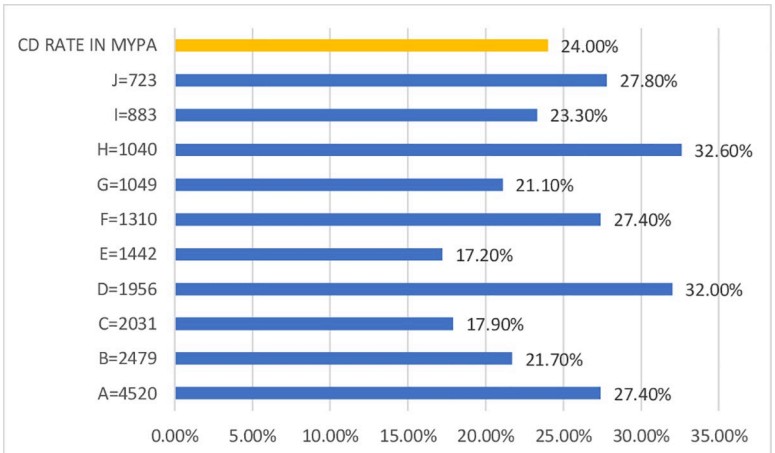

**Fig 1. Cesarean delivery (CD) in each hospital of the MYPA perinatal network.** The CD rates were significantly different between each hospital (p<0,001). Hospital A and B are second level maternities.

nulliparas and paras with no previous CD, with a singleton in cephalic presentation ≥ 37 weeks, with induction of labor or a CD before labor) contained 22.9% of the women, while 10.3% were in group 5 (1,795/17,433; previous CD, singleton in cephalic presentation ≥ 37 weeks). Around 10% of the population was assigned to groups 6 to 10. (Table 2) The CD rates for each group differed significantly between the hospitals (*P* < .001). Fig 2 illustrates the variations in CD rates between hospitals within each TGCS group.

The crude associations of CD occurrence with maternal characteristics and hospital organizational factors are reported in Table 3. CD was significantly associated with a maternal age > 40 years (OR 1.77; 95% CI 1.54–2.01) and with severe preeclampsia (OR 4.2, 95% CI

**Table 1. Maternal characteristics and hospital factors for each group in the Ten Group Classification System (TGCS).**

| Maternal characteristics | Overall | Group 1 | Group 2 | Group 3 | Group 4 | Group 5 | Group 6 | Group 7 | Group 8 | Group 9 | Group 10 |
|---|---|---|---|---|---|---|---|---|---|---|---|
| Maternal age (years): median (IQR) | 31.4 (28–35) | 28.8 (25.9–31.9) | 29.4 (26.1–32.9) | 32.5 (29.4–35.5) | 33.4 (30.3–36.7) | 33.4 (30.2–36.4) | 30.2 (27.4–33.1) | 34.0 (31.0–36.9) | 31.8 (29.0–35.8) | 35,1 (29,3–38,0) | 31.3 (27.8–35.1) |
| age >40 years | 3.9% | 1.6% | 3.4% | 5.8% | 8.6 % | 7.3% | 3% | 9.2% | 10.4% | 14,5% | 6.7% |
| **Woman's level of education** | | | | | | | | | | | |
| University (%) | 61.4% | 65.5% | 62.7% | 58.6% | 57.2% | 56.2% | 72.6% | 51.3% | 66.0% | 72,9% | 53.2% |
| High school (%) | 23.2% | 22.5% | 24.4% | 23.3% | 24.5% | 24.3% | 17.6% | 27.0% | 20.6% | 18,7% | 24.7% |
| Primary and middle school (%) | 15.4% | 12% | 12.9 | 18.1% | 18.3% | 19.5% | 9.8% | 21.7% | 13.4% | 8,3% | 22.1% |
| Mother's occupational activity | 69.7% | 75.6% | 73.4% | 65.4% | 65.2% | 65.3% | 81.8% | 68.3% | 65.4% | 75,0% | 68.2% |
| Severe preeclampsia | 1.1% | 0.2% | 1.5% | 0.5% | 2.1% | 1.5% | 2.7% | 0.7% | 2.2% | 2,0% | 3.7% |
| **Hospital factors** | | | | | | | | | | | |
| Proportion of deliveries in private hospital (%) | 34% | 33% | 38% | 33% | 39% | 39% | 41% | 41% | 21% | 48% | 24% |
| Proportion of deliveries in hospitals with a NICU (%) | 40% | 39.5% | 34.4% | 39.2% | 36.7% | 38.3% | 33.1% | 35.2% | 65.3% | 42,6% | 59% |

NICU: Neonatal intensive care unit.

Qualitative data are given with %.

Quantitative data are given with median and interquartile (IQR).

**Table 2. The relative size, the cesarean delivery rate, and the contribution of each group in Robson's classification to this rate in the MYPA network.**

| | Number of CDs | Number of patients | Group size (%)[1] | Group CD rate (%)[2] | Relative group Contribution to overall CD rate (%)[3] |
|---|---|---|---|---|---|
| Group 1: Nulliparous, single cephalic, ≥37 weeks, in spontaneous labor | 482 | 4300 | 24.7% | 11.2% | 11.5% |
| Group 2: Nulliparous, singleton cephalic, ≥37 weeks, with elective delivery | 752 | 1880 | 10.8% | 40.0% | 18.0% |
| Group 3: Multiparous, singleton cephalic, ≥37 weeks, in spontaneous labor | 173 | 5504 | 31.6% | 3.1% | 4.1% |
| Group 4: Multiparous, singleton cephalic, ≥37 weeks, with elective delivery | 453 | 2123 | 12.2% | 21.3% | 10.8% |
| Group 5: Multiparous, singleton cephalic, ≥37 weeks, with history of CD | 1203 | 1795 | 10.3% | 67.0% | 28.8% |
| Group 6: Nulliparous, singleton, breech, | 281 | 293 | 1.7% | 95.9% | 6.7% |
| Group 7: Multiparous, single breech, | 157 | 196 | 1.1% | 81.1% | 3.8% |
| Group 8: Multiple pregnancies | 313 | 467 | 2.7% | 67% | 7.5% |
| Group 9: Transverse presentation | 46 | 49 | 0.2% | 93.8% | 1.1% |
| Group 10: Singleton, cephalic presentation <37 weeks | 322 | 825 | 4.7% | 39.0% | 7.7% |

1. % = n of women in the group/total N women delivered in the setting x 100.

2. % = n of CS in the group/total N of women in the group x 100.

3. % = n of CS in the group/total N of CS in the setting x 100.

3.01–5.7). Hospital status and the ratio of deliveries per staff member per 24 hours were also associated with CD rates.

Table 4 presents the three stages of the analysis with a multilevel logistic regression model. In the empty model, the hospital-level variance was 0.065 and the MOR 1.27. When the TGCS was included in the model, the variance and the MOR remained similar (variance 0.077, MOR 1.33). However, in the final model, which also included maternal and hospital factors, the interhospital variance decreased substantially by more than 70% (variance 0.019), and the MOR declined to 1.14. In this final model, CD rates were significantly associated with maternal age >40 years (ORa 1.05; 95% CI 1.03–1.06), educational level (ORa 1.20; 95% CI 1.13–1.27), severe preeclampsia (ORa 2.20, 95% CI 2.07–2.33), and private maternity units (ORa 1.72; 95% CI 1.35–2.18). The lack of an NICU, however, was associated with a significantly lower CD rate (ORa 0.74; 95% CI 056–0.99).

After exclusion of births from hospital A, the analysis of the remaining 12,933 births produced similar results and thus confirmed that most of the variation in CD rates among hospitals was related to maternal characteristics and hospital factors.

## Discussion

### Principal findings

In this study, we considered a population in which CD rates varied between hospitals. We developed regression models to quantify the extent to which variations in CD rates between maternity units remained when TGCS was taken into account and how the consideration of women's socioeconomic and clinical characteristics and hospital organization factors might modify this variation. Our analyses showed that patient and hospital factors reduced the inter-hospital variance in CD rates by more than 70%, thereby demonstrating the need to integrate case-mix models when we compare CD rates within groups of the TGCS between hospitals.

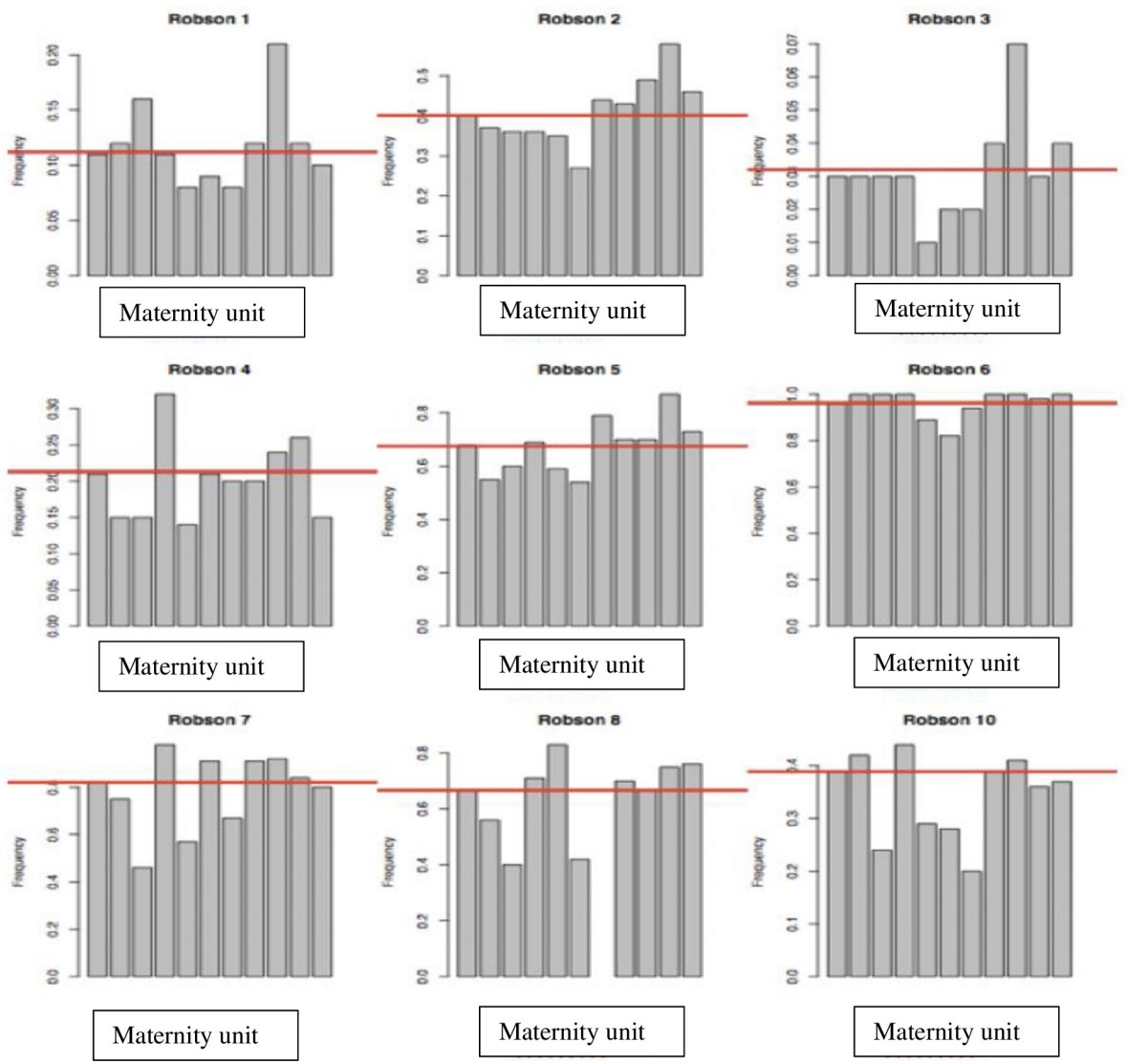

**Fig 2. Cesarean delivery (CD) rate of each hospital.** For each histogram, the first column represents the mean CD rate of the Mypa network, followed by each CD rate of each hospital (A to J). CD rates the Group 9 (fetus in a transverse presentation) are not shown, as the relative size of Group 9 is 0,2% of overall population.

## Implications of the work

As the difficulties in understanding the rise in CD rates worldwide become more generally recognized, the TGCS has emerged as the most appropriate—and the most frequently used—classification for assessing CD rates, as it builds clinically meaningful subgroups of the population of pregnant women and thus makes it possible to analyze variations in CD rates across institutions, countries, and time. Many classification systems are based on indications for CD, such as fetal distress, dystocia, failure to progress, cephalopelvic disproportion, failed induction, macrosomia, or failed trial of labor, but these classifications are limited by ambiguous terminology [21]. Over the last decade, therefore, the TGCS has been adopted for monitoring (and even comparing) CD rates, with the use of specific indicators such as the CD rate within each TGCS group and the relative size of the group (percentage of the total population of women).

**Table 3. Crude odds ratios for cesarean deliveries by maternal socioeconomic and clinical characteristics and hospital organizational factors.**

|  | OR | 95% CI | *P*-value |
|---|---|---|---|
| **Maternal characteristics** | | | |
| **Maternal age** > 40 years | 1.77 | 1.54–2.01 | <0.001 |
| **Educational level** | | | 0.11 |
| University | ref | ref | |
| High school | 1.01 | 0.99–1.08 | |
| Primary and middle school | 1.04 | 1.01–1.07 | |
| **Occupational activity** | 0.97 | 0.90–1.04 | 0.47 |
| **Severe preeclampsia** | 4.16 | 3.01–5.7 | <0.001 |
| **Hospital factor** | | | |
| **Private status** | 1.37 | 1.27–1.47 | <0.001 |
| **No NICU unit** | 0.93 | 0.87–1.04 | 0.06 |
| > 0.92 **deliveries per staff member** per 24h* | 0.89 | 0.83–0.95 | 0.002 |

NICU: Neonatal intensive care unit.

Data are expressed in numbers, percentages, and odds ratios with their 95% confidence intervals

*0.92 deliveries/staff member/24 hours corresponds to the median of deliveries/staff member among MYPA hospitals.

This classification illustrates the variation in CD risk from one group to another and helps to identify groups of women at low and at high risk of CD.

However, while the TCGS is a starting point for describing CD practices in relevant obstetric groups, its utility in accounting for the wide variance of these practices between hospitals is limited. Moreover, use of the TGCS as an audit tool can lead to misinterpretations of CD rate variations between hospitals. Most studies evaluating the TGCS do not describe maternal characteristics, or the aspects of hospital organization that may influence CD rates [29–31]. For

**Table 4. Measures of associations between individual/hospital characteristics and cesarean delivery (CD) and measures of variability of cesarean delivery rates in the MYPA network, France, 2014, obtained from multilevel logistic models.**

|  | Empty model | Model using TGCS | Model using TGCS plus maternal characteristics and hospital factors |
|---|---|---|---|
| **Measures of association with CD occurrence (OR, 95% CI)** | | | |
| Maternal age >40 years | | | 1.05(1.03–1.06) |
| Educational level | | | 1.20 (1.13–1.27) |
| Severe preeclampsia | | | 2.20 (2.07–2.33) |
| Private maternity | | | 1.72 (1.35–2.18) |
| No NICU unit | | | 0.74 (0.56–0.99) |
| >0.92 deliveries/staff member/24h* | | | 0.86 (0.56–1.16) |
| **Measures of variability of CD rates** | | | |
| Area level variance | 0.065 | 0.077 | 0.019 |
| Proportional change in variance | | -18% | 71% |
| MOR | 1.27 | 1.30 | 1.14 |

NICU: Neonatal intensive care unit, OR: Odds ratio, 95%CI: 95% confidence interval.

ICC, intraclass correlation; MOR, median odds ratio.

0.92 deliveries/staff member/24 hours corresponds to the median of deliveries/staff member among MYPA hospitals.

example, we observed that the CD rate was significantly lower in hospitals without NICUs. Similarly, Le Ray et al. analyzed the influence of the maternity unit's perinatal care level on the rate of intrapartum cesarean delivery among women with low-risk pregnancies and reported that maternity units with NICUs that manage high-risk pregnancies have higher rates of cesareans during labor for their population of nulliparas at low risk than do facilities that deal mainly with low-risk pregnancies. They identified the maternity unit level as the only structural factor that was a significant risk factor for cesarean delivery during labor. Le Ray et al. suggested several hypotheses to explain the differences in these cesarean rates as a function of level of perinatal care: Perhaps, in high-risk facilities, physicians are more likely to expect problems and may encourage the use of cesarean deliveries while in low-risk facilities, midwives are less likely to expect problems and may thus encourage vaginal delivery, which is important because support for women during labor is essential for successful vaginal delivery. Another hypothesis centers on the organization of care: if the availability of personnel makes it easier to perform a cesarean delivery in a level-3 unit, this might affect obstetric decision-making [32].

Some authors have suggested that the association of the TGCS with maternal factors and prenatal risk factors may provide the most reliable model for exploring this variation. Studying CD rates in seven international tertiary referral hospitals (as well as a regional referral hospital and the Iceland 2005 birth cohort), Brennan et al. have suggested that the variation of CD rates among nulliparous and parous women in spontaneous labor with singletons in cephalic presentation (TGCS groups 1 and 3) might be related to maternal clinical, social, and demographic risk factors, which the TGCS does not consider [12]. The findings of Colais et al. support this hypothesis, in demonstrating that the residual variability in CDs is explained by clinical, social, and demographic variables and that the TGCS classification does not suffice to eliminate case-mix differences [33]. Pasko et al. also confirmed this point recently, in their estimates of the contributions of the characteristics of patients, healthcare providers, and hospitals to the variation in the frequency of nulliparous, term, singleton, vertex CDs [34]. In a cohort of 115,502 women, they observed that patient characteristics accounted for 24% of the variation in this group, while healthcare provider-hospital characteristics were not significantly associated with CD frequency. These findings, reinforced by ours, indicate that understanding CD rates requires recording not only clinical/obstetric factors, but also factors related to maternity organizational factors (ratio of deliveries/staff, ratio of deliveries/delivery room, number per 24 hours of inhouse attending obstetricians, anesthesiologists, and pediatricians). The third paper of a series on optimizing CD use in the *Lancet* illustrated the components affecting CD frequency in a schematic diagram in which the three outer rings represent the layers of complexity of these factors [35]. We agree that a classification requiring such data is time-consuming and complex and that in the end we still need to compare CD rates. The TGCS is an appropriate classification for highlighting differences in these rates, but we have demonstrated why comparisons of CD rates, even in TGCS groups, must be interpreted carefully.

## Strengths and limitations

The main strength of our study lies in its large population-based sample size, multicenter nature, and prospective design, including the collection of numerous data items that make it possible to adjust for a large set of confounding factors. The maternal clinical, social, and demographic data were prospectively recorded in the database by a research midwife and enabled an exclusion rate for missing data of less than 1%.

Our study used case-mix models, which provide a relevant statistical approach for comparing CD rates across hospitals and are suggested for assessing CD rates as institutional quality

indicators. These models help us to understand the determinants of the variability of health-care quality indicators such as CD rates and thus to understand their crude rates and risk ratios better. Precisely because outcomes may depend on preexisting patient characteristics, simply measuring CD rates may not provide useful insight into quality of care. These models have been widely employed in such clinical disciplines as cardiothoracic surgery, but not yet for evaluating obstetric outcomes [36, 37]. Nonetheless Main et al. have suggested that controlling for maternal age reduces CD rate variability in nulliparous term cephalic singleton pregnancies [38].

We computed the median odds ratio (MOR), which is an epidemiologically more suitable option for obtaining measures of variance in logistic regression. It is not statistically dependent on the prevalence of the outcome and allows area-level variance to be expressed on the well-known OR scale. Therefore, it allows comparison of area-level variations (here the hospital level) with the impact of specific factors [28].

Limitations of this study should be noted. Even though this multilevel model may be generalizable to other studies, the variation of CD rates associated with maternal and hospital factors in our study should be interpreted with caution, as these factors might vary across populations. Furthermore, more than 25% of deliveries occurred in a single hospital, which was the only university hospital of the perinatal network. However, findings remained unchanged after its exclusion in a sensitivity analysis.

We had very little information on other maternal clinical characteristics, specifically status for diabetes mellitus or fetal growth restriction (as we did not have access to prenatal ultrasound findings), which are also important variables to consider in interpreting TGCS results. The CoNaissance program does not collect the ultrasound findings, and although we had the birth weight, we were not able to determine if the fetus was considered either small-for-gestational age or a growth-restricted fetus. The variable for diabetes was missing in more than 50% of cases, and we were not able to differentiate gestational diabetes mellitus treated with or without insulin from one another, or from preexisting (non-pregnancy-related) diabetes [39]. Moreover, there is no evidence that gestational diabetes is associated with a higher risk of CD [40]. Finally, this study does not consider maternal body mass index (BMI) in the risk adjustment, as unfortunately it was not recorded in birth certificates. The well-known association of obesity with higher CD risk means that it might well be a factor of variability in CD rates between hospitals [41]. We suppose that adding these maternal and obstetric risk factors to the final model (which uses maternal characteristics, hospital factors, and TGCS) would probably decrease the area-level variance of the model and therefore decrease a little bit more the amount of variation explained by each hospital.

## Conclusions

The Ten Group Classification System is an easy tool to implement for auditing CD and for highlighting differences in CD rates among maternity units. However, maternal characteristics and hospital organizational factors are needed before comparing CD rates within groups of the TGCS.

## Supporting information

**S1 File.**
(XLTX)

## Author Contributions

**Conceptualization:** Thibaud Quibel.

**Data curation:** Camille Bouyer.

**Formal analysis:** Thibaud Quibel.

**Investigation:** Thibaud Quibel.

**Methodology:** Thibaud Quibel, Patrick Rozenberg, Jean Bouyer.

**Writing – review & editing:** Patrick Rozenberg, Jean Bouyer.

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
