## [Decision Letter · Decision Letter 0]

22 Jan 2021

PONE-D-20-34380

Variation between hospital caesarean delivery rates when Robson's classification is considered: an observational study from a French perinatal network

PLOS ONE

Dear Dr. Thibaud Quibel,

Thank you for submitting your manuscript to PLOS ONE. After careful consideration, we feel that it has merit but does not fully meet PLOS ONE’s publication criteria as it currently stands. Therefore, we invite you to submit a revised version of the manuscript that addresses the points raised during the review process.

I would like to thank you for the opportunity to review the manuscript titled “Variation between hospital caesarean delivery rates when Robson's classification is considered: an observational study from a French perinatal network” by Thibaud Quibel, M.D and colleagues.

The aim of the study was to examine the contribution of maternal socioeconomic and clinical characteristics and hospital organizational factors to the variation in cesarean delivery rates when considering Robson’s ten-group classification system. 

This study was a prospective, observational, population-based study including all deliveries after viability (>24 weeks) at the 10 hospitals of the French MYPA perinatal network in the Paris area.

Upon review of the Robson’s criteria and the literature published concerning the topic, I agree with the authors that considerably relevant factors have not been considered when examining this tool to compare variations in cesarean rates. There have been multiple papers that have demonstrated that maternal and hospital/ system organizational factors clearly impact those calculations and cannot be ignored. If this system/tool is used for any quality indicators to effect change, these important factors should be factored into the score. As a result, I find the paper to be truly relevant and applaud the authors for the development of the concept, design and well thought out statistical plan.The methods are well designed, and the statistics are explained and planned appropriately. I appreciate the sensitivity analysis excluding hospital A, which carried 25% of the deliveries and surely would influence the data given that it was a high-risk hospital. I am not surprised from the findings and results, which make complete sense. All of which are compatible with current high quality obstetrical data.I wonder if the authors believe that the lower risks hospitals which did not have a NICU had lower cesarean rates because perhaps those mothers were transferred out? This would be a nice discussion point.I also wonder why diabetes was not available as an obstetrical variable that may influence cesarean rates. As gestational and pregestational diabetes effect a significant proportion of pregnant women, that variable is hard to leave out. It is listed as a limitation, but the authors do not explain why that variable was left out.With respect to institutional factors, I believe that private and public health and midwife to MD ratios are extremely relevant and appreciate those results. They make complete sense. These results speak to the experience of the providers and the true level of acuity each hospital is comfortable with or equipped to handle. Line 281-4 imply that it is more preference base (financial incentives, fear of litigation, etc) when in fact, these providers may not feel that way but perhaps, the hospitals are not structured and approved for certain medical conditions and obstetrical complications. Personnel and finance are vastly different for a level 1 vs. level 3 hospital. I think those lines can be rewritten to reflect professional objective measures.

This is an excellent paper and I recommend approval, however if possible, if the authors could add the diabetes data. If not, please explain why data is not available more clearly in the discussion. In addition, a brief rewrite of lines 281-4 to reduce subjectivity and last, if the lack of NICU and lower cesarean rate data could be explained in the discussion as well.

We look forward to receiving your revised manuscript.

Kind regards,

Alireza Abdollah Shamshirsaz

Academic Editor

PLOS ONE

Journal Requirements:

2. In your ethics statement in the manuscript and in the online submission form, please ensure that you have discussed whether all data/samples were fully anonymized before you accessed them. Specifically, please state whether any of the authors had access to data with identifying information prior to de-identification in the database. Please also state whether the data were collected specifically for this study, or collected routinely and retrospectively reviewed for this study."

3. In the ethics statement in the manuscript and in the online submission form, please provide additional information about the patient records/samples used in your study, including a) the date range (month and year) during which patients' medical records/samples were accessed; b) the date range (month and year) during which patients whose medical records/samples were selected for this study sought treatment.

Reviewers' comments:

Reviewer's Responses to Questions

**Comments to the Author**

1. Is the manuscript technically sound, and do the data support the conclusions?

Reviewer #1: Yes

2. Has the statistical analysis been performed appropriately and rigorously? 

Reviewer #1: Yes

3. Have the authors made all data underlying the findings in their manuscript fully available?

Reviewer #1: Yes

4. Is the manuscript presented in an intelligible fashion and written in standard English?

Reviewer #1: Yes

5. Review Comments to the Author

Reviewer #1: I have no comments concerning quality of project. THe manuscript is well written and the authors have disclosed all data in a clear fashion. The statistics are well thought out and I have no problems with the design

6. PLOS authors have the option to publish the peer review history of their article (what does this mean?). If published, this will include your full peer review and any attached files.

Reviewer #1: No

---

## [Author Response · Author response to Decision Letter 0]

28 Mar 2021

Dear Editor,

I would like to thank you for the opportunity to review our manuscript and to thank the Reviewer for his/her constructive comments.

The aim of the study was to examine the contribution of maternal socioeconomic and clinical characteristics and hospital organizational factors to the variation in cesarean delivery rates when considering Robson’s ten-group classification system. 

This study was a prospective, observational, population-based study including all deliveries after viability (>24 weeks) at the 10 hospitals of the French MYPA perinatal network in the Paris area.

1) Upon review of the Robson’s criteria and the literature published concerning the topic, I agree with the authors that considerably relevant factors have not been considered when examining this tool to compare variations in cesarean rates. There have been multiple papers that have demonstrated that maternal and hospital/ system organizational factors clearly impact those calculations and cannot be ignored. If this system/tool is used for any quality indicators to effect change, these important factors should be factored into the score. As a result, I find the paper to be truly relevant and applaud the authors for the development of the concept, design and well thought out statistical plan.

2) The methods are well designed, and the statistics are explained and planned appropriately. I appreciate the sensitivity analysis excluding hospital A, which carried 25% of the deliveries and surely would influence the data given that it was a high-risk hospital. I am not surprised from the findings and results, which make complete sense. All of which are compatible with current high quality obstetrical data.

We thank the reviewer for his comments.

3) I wonder if the authors believe that the lower risks hospitals which did not have a NICU had lower cesarean rates because perhaps those mothers were transferred out? This would be a nice discussion point.

Author response to comment:

We thank you very much for this comment, precisely because we do not think that the reason that maternity units without NICUs had lower caesarean rates was those mothers were transferred out. 

In a report with results similar to our findings, Le Ray et al. (Le Ray C, Carayol M, Zeitlin J, Bréart G, Goffinet F; PREMODA Study Group. Level of perinatal care of the maternity unit and rate of cesarean in low-risk nulliparas. Obstet Gynecol. 2006;107:1269-77) analyzed the influence of the maternity unit's level of perinatal care on the intrapartum cesarean rate among women with low-risk pregnancies and reported that maternity units with NICUs- that manage high-risk pregnancies have higher rates of cesareans during labor for their population of nulliparas at low risk than do facilities that deal mainly with low-risk pregnancies. The level of the maternity unit was the only structural factor identified in this study as a significant risk factor for cesarean during labor. Le Ray et al. suggested several hypotheses to explain the differences in these cesarean rates as a function of level of perinatal care: notably, in high-risk facilities, physicians are more likely to expect problems and may encourage the use of cesarean deliveries, while in low-risk facilities, midwives are less likely to expect problems and thus may encourage vaginal delivery, which is important because support for women during labor is essential for the success of vaginal delivery. Another hypothesis centers on the organization of care: if the availability of personnel makes it easier to perform a cesarean delivery in a unit with a NICU , this might affect obstetric decision-making.

This point is now discussed in the Discussion section on p.18. lines 265-279

4) I also wonder why diabetes was not available as an obstetrical variable that may influence cesarean rates. As gestational and pregestational diabetes effect a significant proportion of pregnant women, that variable is hard to leave out. It is listed as a limitation, but the authors do not explain why that variable was left out.

Author response to comment:

We thank the Reviewer for this comment. Indeed, diabetes was a variable of the first certificate. However, we decided to not include this data for two reasons: 

- First, this variable was often left empty and diabetes was noticed in less than 5% of cases which raises the question of the validity of its data. 

- Second, we were not able to distinguish gestational diabetes with or without insulin from preexisting diabetes. Most cases of diabetes during pregnancies are related to gestational diabetes without insulin, which are not associated with an increased risk of cesarean deliveries. Moreover, when diabetes with insulin is required for a gestational diabetes mellitus, there is no evidence that these pregnancies are associated with an increased risk for CD (Treatments for women with gestational diabetes mellitus: an overview of Cochrane systematic reviews)

For these two reasons, we think that this variable should not be added in our model. Therefore, we have added the following comment: 

The CoNaissance program does not collect the ultrasound findings, and although we had the birth weight, we were not able to determine if the fetus was considered either small-for-gestational age or a growth-restricted fetus. The variable for diabetes was missing in more than 50% of cases, and we were unable to differentiate gestational diabetes mellitus treated with or without insulin from one another or from preexisting diabetes. Moreover, there is no evidence that gestational diabetes is associated with a higher risk of CD. 

This comment has been added in the discussion section, page 21, lines 339-345.

5) With respect to institutional factors, I believe that private and public health and midwife to MD ratios are extremely relevant and appreciate those results. They make complete sense. These results speak to the experience of the providers and the true level of acuity each hospital is comfortable with or equipped to handle. Line 281-4 imply that it is more preference base (financial incentives, fear of litigation, etc) when in fact, these providers may not feel that way but perhaps, the hospitals are not structured and approved for certain medical conditions and obstetrical complications. Personnel and finance are vastly different for a level 1 vs. level 3 hospital. I think those lines can be rewritten to reflect professional objective measures.

Author response to comment:

Once more, we really appreciate the reviewer’s comment. We were referring to a recent article in the Lancet, which illustrates the difficulties of understanding and comparing CD rate, because there are so many overlapping levels. Our study was not able to illustrate all these levels, but we thought that a delivery/staff ratio was helpful for illustrating an organizational factor that appears to affect CD rates. 

We have rewritten these sentences as follows: 

These findings, reinforced by ours, indicate that understanding CD rates requires recording not only clinical/obstetric factors, but also factors related to maternity organizational factors (ratio of deliveries/staff, ratio of deliveries/delivery room, number per 24 h of inhouse attending obstetricians, anesthesiologists, and pediatricians). 

These sentences were modified on page 19, lines 295-298.

This is an excellent paper and I recommend approval, however if possible, if the authors could add the diabetes data. If not, please explain why data is not available more clearly in the discussion. In addition, a brief rewrite of lines 281-4 to reduce subjectivity and last, if the lack of NICU and lower cesarean rate data could be explained in the discussion as well. 

We thank the Reviewer.

More, we added in the manuscript modifications that were requested from the editor.

We made modifications which should meet PLOS ONE’s style requirements.

2. In your ethics statement in the manuscript and in the online submission form, please ensure that you have discussed whether all data/samples were fully anonymized before you accessed them. Specifically, please state whether any of the authors had access to data with identifying information prior to de-identification in the database. Please also state whether the data were collected specifically for this study, or collected routinely and retrospectively reviewed for this study."

We have rewritten these sentences as follows: 

Data are extracted from the CoNaissance 78 program, which was created in 2008 to monitor maternal and perinatal morbidity and mortality in the perinatal network. This dataset contains all births in the district with fully anonymized demographic characteristics and medical information about each pregnancy and delivery, as well as about maternal, fetal, and neonatal health. Therefore, any of the authors had access to data with identifying information prior to de-identification in the database.

This is written on page 7, lines 108-10

 3. In the ethics statement in the manuscript and in the online submission form, please provide additional information about the patient records/samples used in your study, including a) the date range (month and year) during which patients' medical records/samples were accessed; b) the date range (month and year) during which patients whose medical records/samples were selected for this study sought treatment.

Similarly, we provided additional information about the patient records used in this study, notably the date range:

Since its creation, data are continuously recorded from two health certificates completed for each birth in the network at the hospital of delivery.

This is written on page 7, lines 113-114.

The date range of this study started on the first January 2014 and ended the 31st December 2014. This was precised on page 7 line 97-98

Sincerely yours,

Thibaud Quibel

---

## [Editor Report · Decision Letter 1]

21 Apr 2021

Variation between hospital caesarean delivery rates when Robson's classification is considered: an observational study from a French perinatal network

PONE-D-20-34380R1

Dear Dr. Quibel,

We’re pleased to inform you that your manuscript has been judged scientifically suitable for publication and will be formally accepted for publication once it meets all outstanding technical requirements.

Kind regards,

Alireza Abdollah Shamshirsaz

Academic Editor

PLOS ONE
---

## [Editor Report · Acceptance letter]

11 Aug 2021

PONE-D-20-34380R1 

Variation between hospital caesarean delivery rates when Robson's classification is considered: an observational study from a French perinatal network 

Dear Dr. Quibel:

I'm pleased to inform you that your manuscript has been deemed suitable for publication in PLOS ONE. Congratulations! Your manuscript is now with our production department. 

Kind regards, 

on behalf of

Dr. Alireza Abdollah Shamshirsaz 

Academic Editor

PLOS ONE